# Ellipticity dependence of high-harmonic generation in solids originating from coupled intraband and interband dynamics

Nicolas Tancogne-Dejean [1,2], Oliver D. Mücke [3,4], Franz X. Kärtner[3,4,5] & Angel Rubio [1,2,3,5]

The strong ellipticity dependence of high-harmonic generation (HHG) in gases enables numerous experimental techniques that are nowadays routinely used, for instance, to create isolated attosecond pulses. Extending such techniques to solids requires a fundamental understanding of the microscopic mechanism of HHG. Here we use first-principles simulations within a time-dependent density-functional framework and show how intraband and interband mechanisms are strongly and differently affected by the ellipticity of the driving laser field. The complex interplay between intraband and interband effects can be used to tune and improve harmonic emission in solids. In particular, we show that the high-harmonic plateau can be extended by as much as 30% using a finite ellipticity of the driving field. We furthermore demonstrate the possibility to generate, from single circularly polarized drivers, circularly polarized harmonics. Our work shows that ellipticity provides an additional knob to experimentally optimize HHG in solids.

[1] Max Planck Institute for the Structure and Dynamics of Matter, Luruper Chaussee 149, 22761 Hamburg, Germany. [2] European Theoretical Spectroscopy Facility (ETSF), Luruper Chaussee 149, 22761 Hamburg, Germany. [3] Center for Free-Electron Laser Science CFEL, Deutsches Elektronen-Synchrotron DESY, Notkestraße 85, 22607 Hamburg, Germany. [4] The Hamburg Center for Ultrafast Imaging, Luruper Chaussee 149, 22761 Hamburg, Germany. [5] Physics Department, University of Hamburg, Luruper Chaussee 149, 22761 Hamburg, Germany. Correspondence and requests for materials should be addressed to N.T.-D. (email: nicolas.tancogne-dejean@mpsd.mpg.de) or to A.R. (email: angel.rubio@mpsd.mpg.de)

aking advantage of the polarization state of light pulses has recently opened up tremendous, unprecedented opportunities for investigating and controlling strong-field interactions in atomic and molecular gases. The polarization degree of freedom is not only important for studying fundamental physical aspects of light–matter interactions, but a time-varying polarization state[1, 2] underlies numerous spectroscopy and coherent control techniques in attoscience, and it is technologically relevant for tabletop high-harmonic sources in the extreme ultraviolet (XUV) and soft X-ray spectral regions.

For example, in atomic and molecular gases, attosecond recollision-based physical processes, such as laser-induced electron diffraction[3], nonsequential double ionization[4], above-threshold ionization[5, 6], and high-harmonic generation (HHG)[6, 7], are extremely sensitive to small deviations from linear polarization due to the resulting lateral displacement of the returning electron wavepacket with respect to the parent ion (as nicely accounted for by the standard recollision model of strong-field physics[8–10]). The ellipticity dependence of HHG was recently used to probe the molecular chirality on a sub-femtosecond electronic timescale[11]. More technologically, this ellipticity sensitivity has been successfully exploited in several gating schemes for the production of isolated attosecond XUV pulses, e.g., by polarization gating[12] and (generalized) double optical gating[13, 14].

Coherent steering of the electron wavepacket in a two-dimensional plane using orthogonally polarized two-color laser fields allows to measure the tunnel ionization time and recollision time[15], as well as probing the parent ion with the electron returning under different angles with attosecond precision[16], which brings intriguing applications in the tomography of atomic or molecular wavefunctions[17]. Even more elaborate schemes using counter-rotating circularly polarized laser fields at different wavelengths has lead to the recent demonstration of bright circularly polarized soft X-ray high-harmonic sources with fascinating spectroscopic applications in magnetic materials using X-ray magnetic circular dichroism (XMCD) (see, e.g., refs. [18–21], and earlier works cited therein).

The use of circularly polarized fields opens the door to producing vortex-shaped photoelectron momentum distributions[22] as well as studying spin-polarized electrons created by non-adiabatic tunneling[23–26], attosecond control of spin-resolved recollision dynamics[26, 27], and investigating ionization dynamics from atoms and molecules via angular streaking (atto-clock)[28–31] using cold target recoil ion momentum spectroscopy (COLTRIMS).

From the above, it is clear that the driver field's ellipticity for strong-field interactions in gases has opened up a plethora of interesting physical phenomena to explore. In contrast, the role of ellipticity in strong-field interactions in solids remains so far largely unexplored, thus hampering the possibility to exploit or extend some of the above-mentioned experimental techniques to solid-state devices.

The first experimental investigation of the impact of the ellipticity of the driving laser field on HHG from bulk ZnO[32] showed that the emitted harmonics are less sensitive to ellipticity than harmonics originating from gases. However, like in atoms and molecules, circularly polarized light suppresses HHG from this material[32]. Solving the semiconductor Bloch equations for a two-band model for ZnO showed that the harmonic yield monotonically decreases with a Gaussian profile with increasing ellipticity[33]. Such an atomic-like monotonic decrease of the harmonic yield with increasing driving field ellipticity was recently observed experimentally also from rare-earth solids[34] and monolayer MoS$_2$[35]. However, a later work on bulk MgO[36] reported that, unlike in gases, HHG from bulk crystals can exhibit strongly anisotropic ellipticity profiles. The authors showed that the maximal harmonic yield can, in some cases, be reached not for linear polarization, but for a finite value of the ellipticity $\epsilon$. Their experimental results also revealed that, counter-intuitive to previous belief, circularly polarized driver pulses do not always prohibit harmonic generation from bulk crystals.

In order to explain the strongly anisotropic ellipticity dependence of HHG in MgO, You et al. proposed a model based on classical real-space trajectories in a two-dimensional one-band model including scattering from neighboring atomic sites. However, their simple picture of pure intraband dynamics is physically incompatible with real-space classical trajectories: In fact, the adiabatic evolution within one band in momentum space (Bloch oscillations) corresponds to a Wannier–Stark localization in real space[37], for which electrons localize at different atomic sites of the crystal[38], as experimentally observed in semiconductor super-lattices[39]. The possibility of maximal harmonic yield at finite ellipticity was proposed for solids, in the regime of semi-metallization of the crystal[40]. As this semi-metallization regime occurs at a much higher intensity than used in the HHG experiments in solids so far, it cannot explain the experimental results of ref. [36].

Here, we investigate, using an ab initio approach based on time-dependent density-functional theory (TDDFT)[41, 42], the role of ellipticity in HHG from solids. Simulations are performed for bulk silicon and bulk MgO. We follow the approach we have recently introduced in ref. [43] to describe HHG in solids with full inclusion of electronic band structure and crystal structural effects (see ref. [43] and the "Methods" section for more technical details). In the following, the ellipticity parameter is denoted as $\epsilon$, which varies from −1 (left-handed circular polarization) to 0 (linear polarization) to +1 (right-handed circular polarization). We demonstrate that the complex interplay between intraband and interband effects can be used to tune and improve harmonic emission in solids, for instance extending the high-harmonic plateau by as much as 30% using a finite ellipticity of the driving field. Also, we demonstrate the possibility to generate, from a single circularly polarized driving field, circularly polarized harmonics in cubic Si and MgO with alternating helicity.

## Results

**Influence of the ellipticity of the driving field.** We start by analyzing the ellipticity dependence of HHG in the case of bulk silicon. The vector potential acting on the electrons is given by (atomic units are used throughout this paper)

$$\mathbf{A}(t) = \frac{\sqrt{I_0}c}{\omega}f(t)\left[\frac{1}{\sqrt{1+\epsilon^2}}\cos(\omega t + \phi)\widehat{\mathbf{e}}_x + \frac{\epsilon}{\sqrt{1+\epsilon^2}}\sin(\omega t + \phi)\widehat{\mathbf{e}}_y\right],$$

(1)

where $I_0$ is the peak intensity inside matter, $f(t)$ the (normalized) envelope, $\omega$ the carrier photon frequency, and $\phi$ the carrier-envelope phase (CEP).

We neither account for any electronic dephasing nor propagation effects in our simulations, but we found that the recent experimental ellipticity profiles of HHG in bulk MgO are well reproduced by our theoretical description (see below), showing the reliability of our theoretical description. Surface effects, as well as light-propagation effects and dissipation via phonons are beyond the scope of the present work.

Considering the microscopical mechanism underlying HHG from solids, we note that if the laser field is elliptically polarized with a major axis along a mirror plane of the Brillouin zone (BZ) of the crystal, the left-handed (defined here by negative ellipticity $\epsilon$) and right-handed (positive ellipticity) helicities are equivalent. This is well understood as the HHG mechanism reflects the

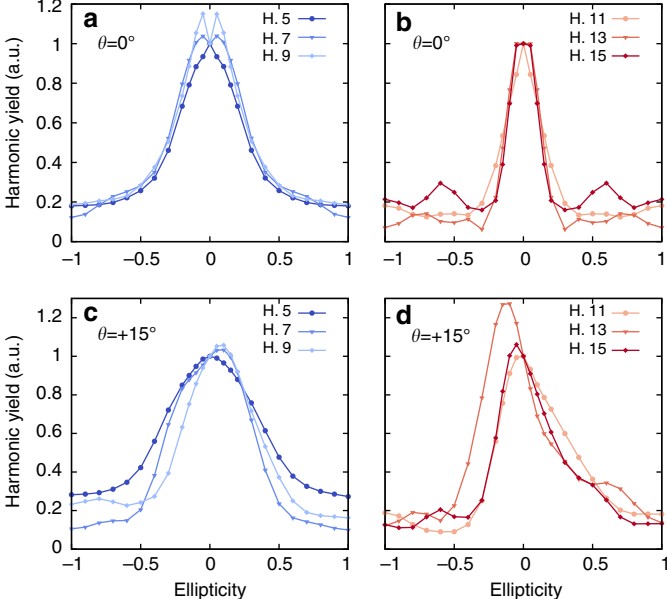

**Fig. 1** Ellipticity profiles of HHG from bulk silicon. Computed ellipticity dependence of the various odd harmonics (5–15) generated from bulk silicon for laser polarization **a**, **b** along the $\overline{\Gamma X}$ direction ($\theta = 0°$), and **c**, **d** for laser polarization rotated by +15° around the [001] crystallographic axis. Two distinct responses are observed for harmonics 5–9 and for harmonics 11–15 (see main text for details)

symmetries of the BZ[43]. Following the same argumentation, if the major axis of the polarization ellipse of the driving field is not aligned with a major axis, we expect an anisotropic profile as left-handed and right-handed helicities will drive electrons into different and non-equivalent regions of the BZ, as experimentally observed recently[36]. Our simulation results, shown in Fig. 1, clearly predict an isotropic ellipticity profile for a laser polarization along the $\overline{\Gamma X}$ direction (Fig. 1a, b), whereas an anisotropic profile is found if the major axis of polarization is rotated by +15° around the [001] crystallographic axis (Fig. 1c, d).

Interestingly, our results show that for the laser polarization along the $\overline{\Gamma X}$ direction (Fig. 1a, b), the harmonics 5–9 exhibit a very similar ellipticity dependence. In contrast, harmonics 11–15 present a different profile but exhibit all a very similar ellipticity dependence. This puts in evidence that the physical interpretation of ref. [36], for which classical real-space trajectories can only lead to the same ellipticity dependence for all the emitted harmonics, should be revisited, as we will do next here.

HHG from solids originates from two mechanisms, the interband mechanism, which corresponds to direct electron–hole recombination, and the intraband dynamics, in which the carriers are accelerated within the bands by the driver field. Recently, we demonstrated in ref. [43] the possibility to predict spectral regions in the emitted HHG spectra, where the interband contribution is suppressed, from the knowledge of the joint density of states (JDOS). For the same material and laser parameters used here, it was found that harmonics 7 and 9 do not exhibit a clean odd-harmonic peak structure and appear quite noisy, which is consistent with both interband and intraband mechanisms contributing to HHG. On the other hand, the harmonics originating mostly from the intraband mechanism (harmonics 11–15) were found to have a much cleaner spectral structure[43]. From the results displayed in Fig. 1 (and from the ellipticity of the emitted harmonics, see below), we recover the same grouping of harmonics, based on their ellipticity

dependence. This indicates that interband and intraband mechanisms respond differently to ellipticity.

In order to get deeper insight in this interpretation, we have reproduced the same simulations, but for a laser polarization rotated by +15° around the [001] crystallographic axis. Again, our results (see Fig. 1c, d) show that harmonics 7 and 9 behave similarly, as they are all slightly biased toward right-handed helicity, whereas harmonics 11–15 are biased toward left-handed helicity. Note that the harmonic 5 (mostly below bandgap) remains peaked at linear polarization. This clearly shows that these two groups of harmonics do not have the same physical origin, and indicates that the two main microscopic mechanisms responsible for HHG in solids, namely the interband and intraband mechanisms, are affected differently by the ellipticity of the driving field.

Our conclusions are further supported by simulations for bulk MgO (see Fig. 2), which are in qualitative agreement with experiments[36]. We found that low-order harmonics exhibit a completely different ellipticity dependence than higher orders, as presented in Fig. 2e-h. This reflects well the altered interplay between the interband and intraband dynamics, which are differently affected by the ellipticity. It is important to note that even if dephasing and propagation effects are not included here, we still obtain a good agreement with experiments. Propagation effects can affect the harmonic spectra and the harmonic yield[44]. However, as MgO is a cubic isotropic material, the linear absorption only depends on the wavelength but not on the polarization state of the emitted harmonics. Therefore, the harmonic yield of different polarizations will be affected equally by light-propagation effects, irrespective of the driver's and harmonics' ellipticities. Moreover, since the harmonics originate within few nanometers close to the back surface of the crystal, nonlinear effects in the propagation of the emitted harmonics can be neglected. Concerning dephasing effects, which originate from electron–electron and electron–phonon scattering, if they are treated within the relaxation time approximation ($T_2$), the dephasing time is a material property, assumed to be independent of the excitation condition. This is the approximation which has been adopted in most of the HHG papers, even producing a good agreement with experiments[45–48]. Most importantly, assuming a constant dephasing time $T_2$, dephasing will not change the lineshape of the ellipticity profiles.

It is very important to make a close connection to the case of HHG in atoms. Indeed, in some cases[49, 50] an increase of the harmonic yield for an ellipticity $\epsilon \sim 0.1$ has been observed, and it was proposed that these harmonics could originate from bound–bound transitions[50]. In the case of solids, this scenario would correspond to interband transitions. In the case of silicon (Fig. 1a), we observe such an increase for the harmonics 7 and 9, which have both interband and intraband contributions for our excitation conditions[43]. Harmonics 11–15, which are mainly originating from intraband contributions, do not exhibit such increase. This is just an indication of the role of interband transitions that would require further work to see if it is a general effect or specific of this system.

The fact that the two mechanisms depend differently on the ellipticity of the driving electric field can be understood as follows: In the case of harmonic emission from the interband mechanism, the emission only depends on optical transitions between available energy levels. In the simplified ideal case of emission of harmonics by a pure interband mechanism, the electrons only perform transitions, independently of how they are steered by the laser field in momentum space. This means that left-handed and right-handed elliptic polarizations should not contribute differently to the interband mechanism, as in both cases the field strength, and thus the excitation of electrons, is identical. On the

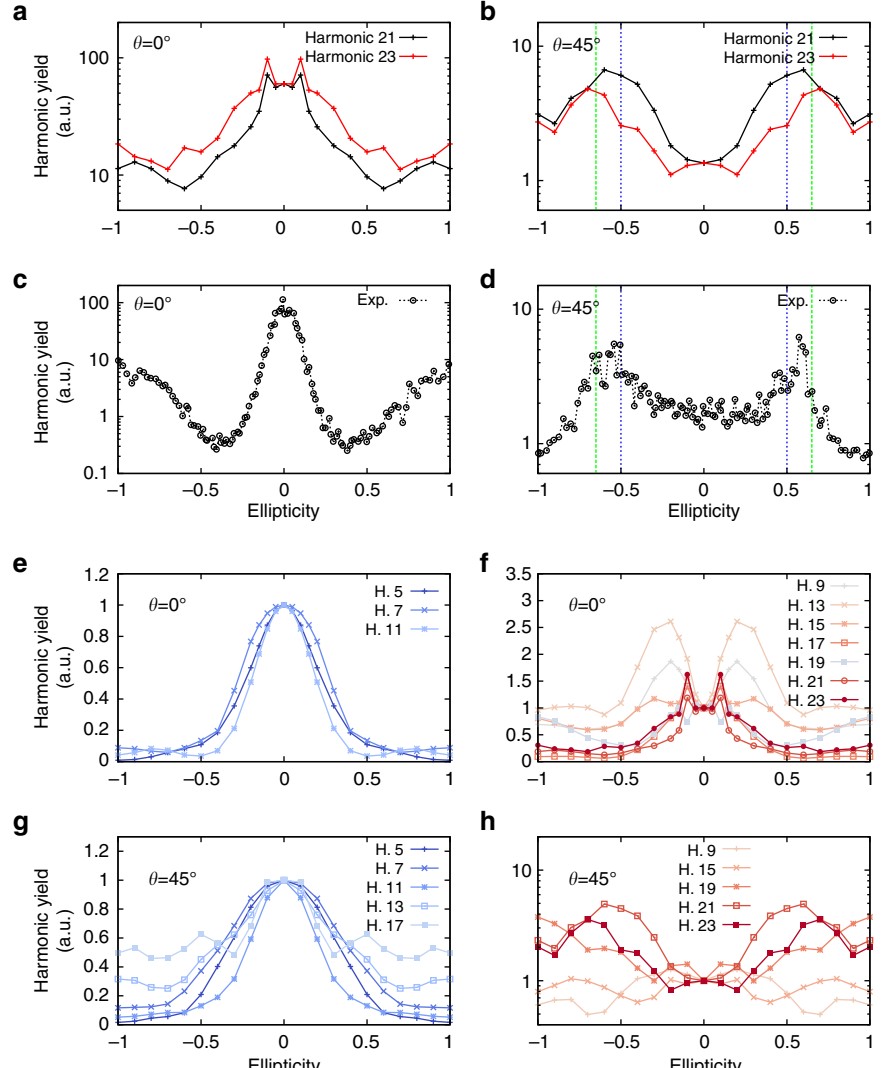

**Fig. 2** Ellipticity dependence of the harmonic yield for different harmonics of bulk MgO. **a**–**d** Computed ellipticity dependence of the harmonics 21 (*black lines*) and 23 (*red lines*) generated from bulk MgO (**a**, **b**) vs. the experimentally observed ellipticity dependence of harmonic 19 (**c**, **d**) taken from ref. [36]. As in Fig. 1, $\theta = 0°$ corresponds to the $\overline{\Gamma X}$ direction. *Vertical lines* in **b**, **d** showing the case of the Mg–O direction ($\theta = 45°$) indicate the positions of $\epsilon = \pm 0.5$ and $\epsilon = \pm 0.65$. **e**–**h** Ellipticity dependence of various harmonics, for the major axis of the polarization ellipse at $\theta = 0°$ (**e**, **f**) and $\theta = 45°$ (**g**, **h**). The harmonics are divided into a group of mostly atomic-like (**e**, **g**) and non-atomic-like harmonics (**f**, **h**), the latter exhibiting a pronounced increase of the harmonic yield at non-zero values of ellipticity

other hand, the intraband mechanism directly probes the conduction bands' dispersions (i.e., the group velocity of the electron wavepacket in momentum space). Moreover, any avoided electronic crossing in the band structure can result in diabatic electronic dynamics, whose concomitant harmonic emission depends of how electrons are driven to this avoided crossing. For HHG in solids, the complex interplay between interband and intraband mechanisms leads to a different weight for each harmonic[43], and it is therefore natural to find a variety of ellipticity profiles for different harmonic orders from the same crystal, as shown in particular in Fig. 2e–h.

**Subcycle dynamics of excited electrons**. In ref. [43], we showed that the harmonic yield is enhanced when the interband mechanism is suppressed by band-structure effects. We now propose to take advantage of the ellipticity of the laser field to drive the electrons into a specific region of the BZ to enhance HHG. In order to demonstrate the driving of the electron wavepacket in momentum space by the external laser in a real

material, we computed the dynamics of the excited electrons, resolved in momentum space. Our results presented in Fig. 3 show that electrons are excited, starting as soon as the field reaches a critical value (for which a sufficient fraction of valence electrons can be excited to the conduction bands). The electron wavepacket is then subsequently accelerated by the vector potential of the applied laser, indicated by the *black arrow*. The region the electrons explore is dictated by the instantaneous value of the vector potential. Moreover, the various snapshots of the excited electrons show a complex modulation at a subcycle timescale, due to the complex band structure of silicon, which results in many conduction bands being involved in the dynamics (A video of the full-time evolution of the momentum-resolved subcycle dynamics of excited states is provided as Supplementary Movie 1).

**Momentum space trajectories**. It might of course be tempting to interpret the dynamics of the electron wavepacket in terms of **k**-space trajectories using the so-called acceleration theorem[37].

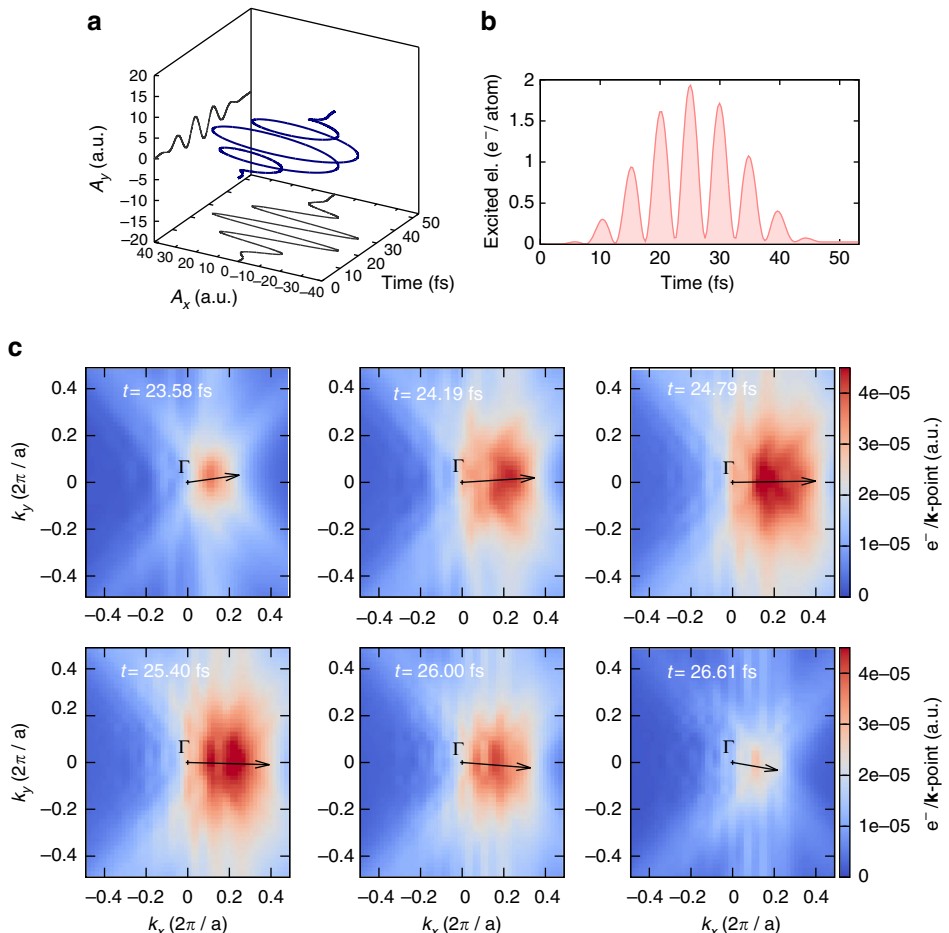

**Fig. 3** TDDFT simulations of the subcycle dynamics of the excited electrons. Simulations are performed for bulk Si, around the maximum of the laser vector potential, in the $k_z = 0$ plane. **a** The left-handed applied vector potential, with an ellipticity of $\epsilon = 0.1$. **b** Number of electrons excited to the conduction bands during the laser pulse. **c** Momentum space resolved subcycle dynamics of the excited electrons. The *black arrow* indicates the direction and strength of the applied vector potential. The number of excited electrons (displayed as colormap) is computed by projecting the time-evolved wavefunctions on the ground-state Kohn–Sham wavefunctions (see "Methods" section)

This has been done, for instance, in refs. [45, 51, 52] for few-band models or analytical potentials. However, due to the complexity of the band structure of even simple semiconductors, such as silicon, involving many bands close to the bandgap, the validity of such a simple analysis must be scrutinized. In particular, the underlying adiabatic approximation implies that interband transitions, level crossings and avoided crossings are neglected.

The acceleration theorem states that[37], under the approximation of an adiabatic evolution, the evolution of the electron wavepacket momentum $\mathbf{k}_e$ is given by:

$$\frac{d\mathbf{k}_e}{dt} = \mathbf{F}(t), \qquad (2)$$

where $\mathbf{F}(t)$ is the force acting on the electron wavepacket. Neglecting electron–electron and electron–phonon scattering, this force reduces to the driving electric field. More precisely, the above Eq. (2) is only valid if the electron wavepacket remains in the same band, and does not interact with other electrons or phonons. It is therefore clear that it cannot describe a situation where the interband mechanism dominates over the intraband mechanism.

In a typical HHG experiment, electrons are first excited from the valence to the conduction bands during the laser pulse. In particular, the field strength needs to reach a critical value, such

that multiphoton ionization or interband Zener tunneling occur with significant probability[38]. Afterward, depending on its birth time $t_b$ (adopting the language of the three-step model[8–10]), an excited electron wavepacket will be driven along different trajectories in momentum space, assuming that no interband transitions take place. Assuming the validity of the acceleration theorem (i.e., an adiabatic evolution), the trajectory of the center-of-mass of an electron wavepacket is thus given by:

$$\mathbf{k}_e(t) = \mathbf{k}_e(t_b) - \frac{1}{c}\left(\mathbf{A}(t) - \mathbf{A}(t_b)\right), \qquad (3)$$

where $t_b$ is the birth time of the electron wavepacket, i.e., the moment at which it is created. This time accounts for the fact that in HHG experiments reported so far for bulk crystals, the electrons are excited by interband transitions during the laser pulse. In many previous works, however, the electron wavepacket was usually assumed to already exist before the pulse arrives (for $t_b \to -\infty$), then Eq. (3) reads

$$\mathbf{k}_e(t) = \mathbf{k}_e(t_b) - \frac{1}{c}\mathbf{A}(t). \qquad (4)$$

Assuming now, just for the sake of argument and illustration, that an electron wavepacket can be created at any birth time, we obtain for each instant in time $t$ a set of positions $(k_x, k_y)$,

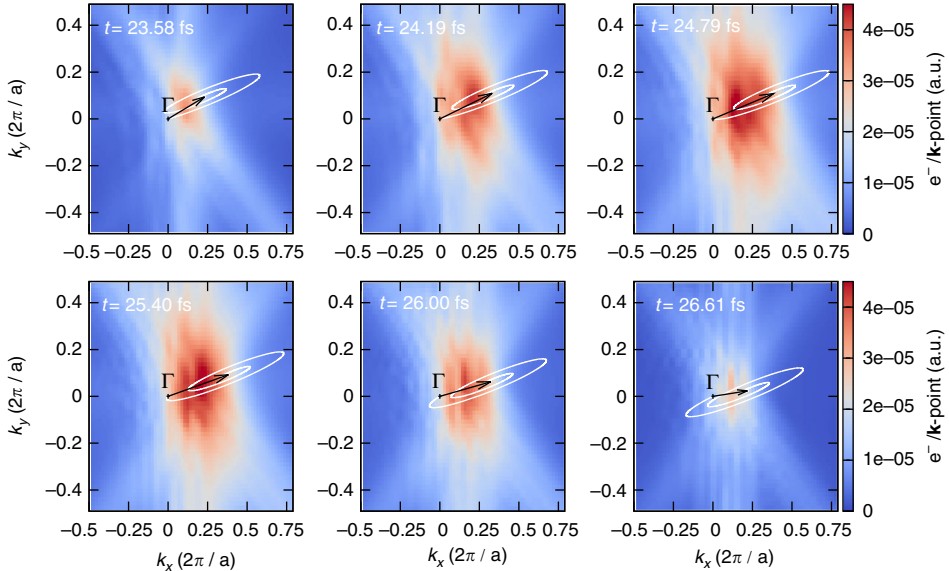

**Fig. 4** Dynamics of the excited electrons in momentum space. Comparison of the time evolution of the excited electron wavepacket in momentum space $N_{ex}(\mathbf{k}; t)$ computed by TDDFT for bulk Si, together with possible positions predicted by the acceleration theorem. The *black arrow* indicates the direction and strength of the vector potential. For each time $t$, the *white curve* corresponds to all possible center-of-mass positions of the wavepacket, for all possible birth times $t_b$ of the wavepacket. The major axis of the polarization ellipse is rotated by $\theta = +15°$ around the [001] crystallographic axis

corresponding to all wavepackets created at all previous birth times $t_b$. As the minimal (direct) bandgap of silicon is located at the $\Gamma$ point, we have $\mathbf{k}_e(t_b) = 0$.

Inspecting the time evolution of the number of excited electrons shown in Fig. 3b, it is clear that under our excitation conditions, most of the excited electrons are virtually excited electrons, whose number returns almost to zero after each half cycle. We also note that a more elaborate model should also take into account a critical value for the electric field $E_c$. Below this value, no significant portion of the valence electrons are really excited into the conduction bands. By choosing such a value, one restricts the values of the birth time to the cases for which $|E(t_b)| > E_c$. Here we do not attempt to propose such an elaborate model, in particular because the critical electric field is not a well-defined quantity, and moreover, such a sophisticated model would only remove some of the possible birth times, hence not changing drastically the conclusions drawn from our simple trajectory analysis. Assuming that all prior times are possible times of birth for the wavepacket, we obtain the trajectories as shown in Fig. 4 for the case of the major axis of the polarization ellipse being rotated by $\theta = +15°$ around the [001] crystal-lographic axis.

In this case, as well as in all cases we investigated, we found that the trajectories obtained from the acceleration theorem agree poorly with our ab initio TDDFT results. Indeed, the acceleration theorem predicts possible positions of the wavepacket in a wider region of the BZ than actually explored by the electrons, according to our ab inito simulations. This shows that neglecting the interband dynamics is not valid for bulk silicon and for our excitation conditions. Overall, this result points toward the breakdown of the simple models used in the literature for explaining HHG from solids in a pure trajectory picture.

**Ellipticity-based HHG cutoff extension**. The energy cutoff of HHG spectra has always been of main importance for techno-logical applications. In solids, this cutoff depends on the max-imum peak of the driving electric field[32], as well as on the polarization direction of the driving field, even in cubic materi-als[43]. We now show that the cutoff energy also depends on the

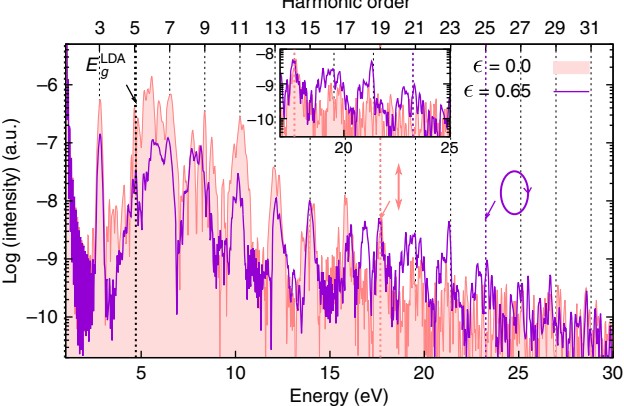

**Fig. 5** Ellipticity-based HHG cutoff extension. Computed HHG spectra from bulk MgO for linearly polarized and elliptically polarized ($\epsilon = 0.65$) driving pulses. In both cases, the major axis of the polarization ellipse along the $\overline{\Gamma K}$. The *dashed lines* indicate the position of the cutoff energy ($E_c$). The inset shows a zoom into the spectral region around the extended energy cutoff

ellipticity of the driving field and that, in contrast to gases, it can even be increased for finite ellipticity in some cases. In order to show the effect of ellipticity on the HHG cutoff, we exploit the case of HHG from bulk MgO with a laser polarization along the Mg–O bond. It was found experimentally in ref. [36], that an ellipticity of $\epsilon = 0.65$ results in an increase of the harmonic yield of bulk MgO by almost one order of magnitude for one of the highest harmonics (19th order). This is well reproduced by our TDDFT simulations (see Fig. 2b).

Our results, reported in Fig. 5, show that when the ellipticity of the laser is changed from linear polarization ($\epsilon = 0$) to the ellipticity that maximizes both experimental and theoretical harmonic yields of the highest harmonics ($\epsilon = 0.65$; see Fig. 2), the cutoff energy for the HHG is increased by up to 30%. Thus, our results clearly show that it is possible to strongly modify and even increase the cutoff energy by changing the ellipticity of the

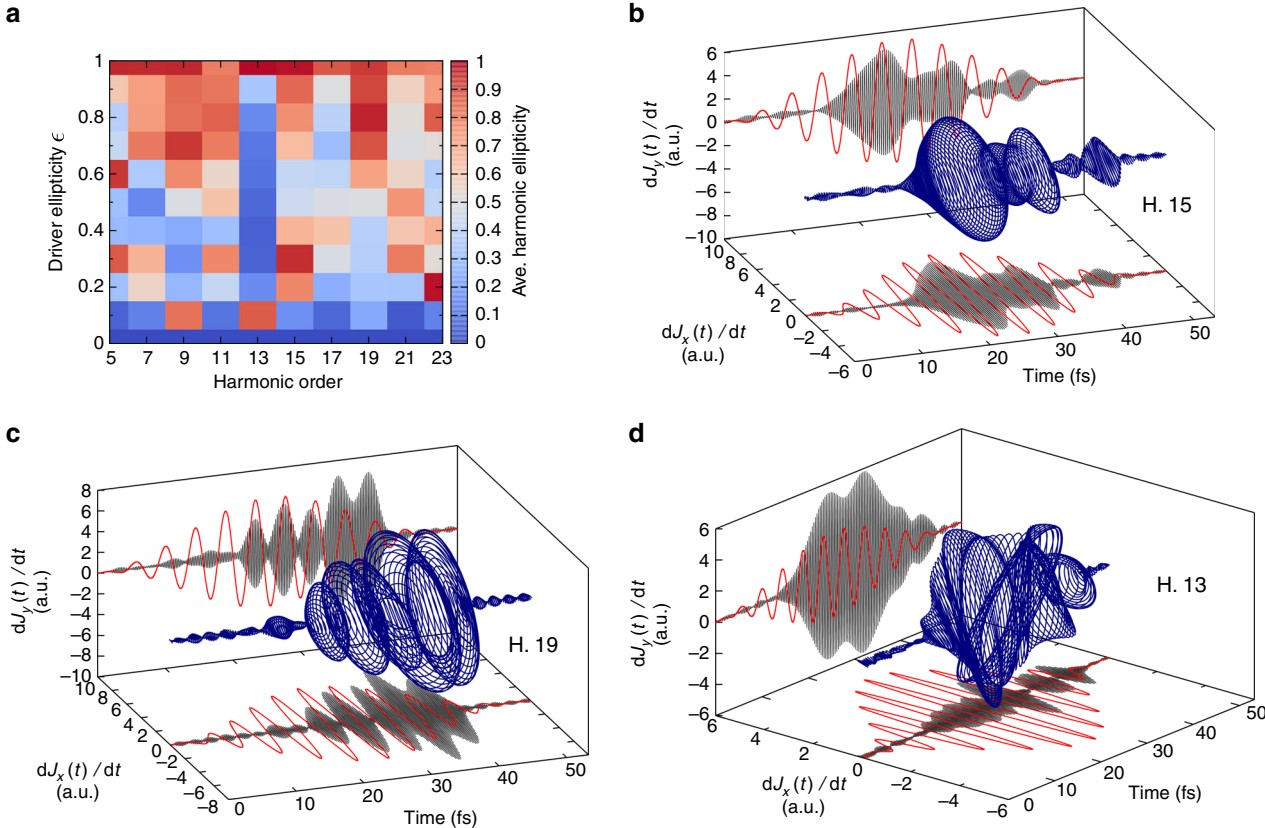

**Fig. 6** Ellipticity of the emitted harmonics for bulk MgO. **a** Calculated average ellipticity of the harmonics emitted for the major axis along the $\overline{\Gamma X}$ direction, vs. the ellipticity of the driving field. **b** Evolution of the time-derivative of the electronic current, bandpass filtered around the 15th harmonic (see "Methods" section), for circular polarization ($\epsilon = 1$). **c** Same as **b**, but for the 19th harmonic. **d** Same as **b** and **c**, but for the 13th harmonic and $\epsilon = 0.6$

incoming laser from linear polarization to some finite ellipticity. This increase of the cutoff is even more impressive, considering that the maximum field at finite ellipticity is a factor $\frac{1}{\sqrt{1+\epsilon^2}}$ (=0.84 for $\epsilon = 0.65$) smaller than the field strength for linear polarization. Therefore, assuming a linear scaling of the cutoff in field strength, we should have found an energy cutoff around 15 eV, i.e., the 15th harmonic. We instead obtain harmonics up to the 25th harmonic from our first-principles simulations.

Our findings highlight that the HHG energy cutoff is not only dictated by the incoming laser field strength but is, in fact, strongly affected by the potential energy landscape felt by the electrons, i.e., the part of the band structure explored by the electrons driven by the strong laser field. The position in momentum space **k** of the electron wavepacket determines its energy, its velocity (via the band dispersion), and its coupling to other bands. As shown in ref. [43], when the JDOS taken from the region of the BZ explored by the electrons is low, the harmonic yield and the cutoff are increased. The cutoff extension can be understood as resulting from the lower JDOS seen by electrons when driven with a finite ellipticity laser, which implies a lower contribution of the interband mechanism in favor of the intraband mechanism, compared to the linearly polarized case[43]. From this perspective, it appears that the laser polarization direction as well as the ellipticity are natural tools to coherently steer electrons inside the BZ, thereby controlling and optimizing the HHG cutoff energy from bulk crystals.

**Ellipticity and helicity of the emitted harmonics**. In ref. [48], it was speculated that it could be possible to generate circularly polarized high-order harmonics from a solid driven by a single-color circularly polarized driving field. It is clear that this

could lead to new and simpler spectroscopy techniques, such as XMCD, for studying magnetic materials[19–21] compared to the bi-color counter-rotating driver fields used in the gas case.

We evaluate now how the average ellipticity of the emitted harmonics (see "Methods" section) depends on the ellipticity of the driving field. Our results, presented for bulk MgO in Fig. 6a, show that even if the ellipticity (averaged over the pulse duration) of the emitted harmonics does not exactly reproduce the ellipticity of the driving field, there is a clear general trend that the ellipticity of the harmonics increases with increasing ellipticity of the driving field. Only the 13th harmonic exhibits an average ellipticity close to zero for all driver ellipticities in Fig. 6a. However, as Fig. 6d reveals, the ellipticity of the 13th harmonic is simply averaging out to zero due to the time-varying rotation of the polarization ellipse.

In order to get more insight, and to check if it is possible to generate circular harmonics with a single circular laser pulse, we computed the evolution of the time-derivative of the electronic current, filtered in frequency around certain harmonics. For a circularly polarized driving field, our results (see Fig. 6b, c) show clearly that the emitted harmonic fields are also mainly circularly polarized. This result demonstrates the possibility to generate circularly polarized high-order harmonics from a single-color circularly polarized driver pulse in solids, opening up the door to future investigations of magnetic materials. This is a major difference to the HHG in gases, where single circularly polarized driver pulses cannot generate harmonics.

A deeper analysis of the results (a video is provided as Supplementary Movie 2) also reveals that the harmonics obtained by the circular driver have alternating helicities, similarly to what has been reported previously in the case of atoms for bi-color

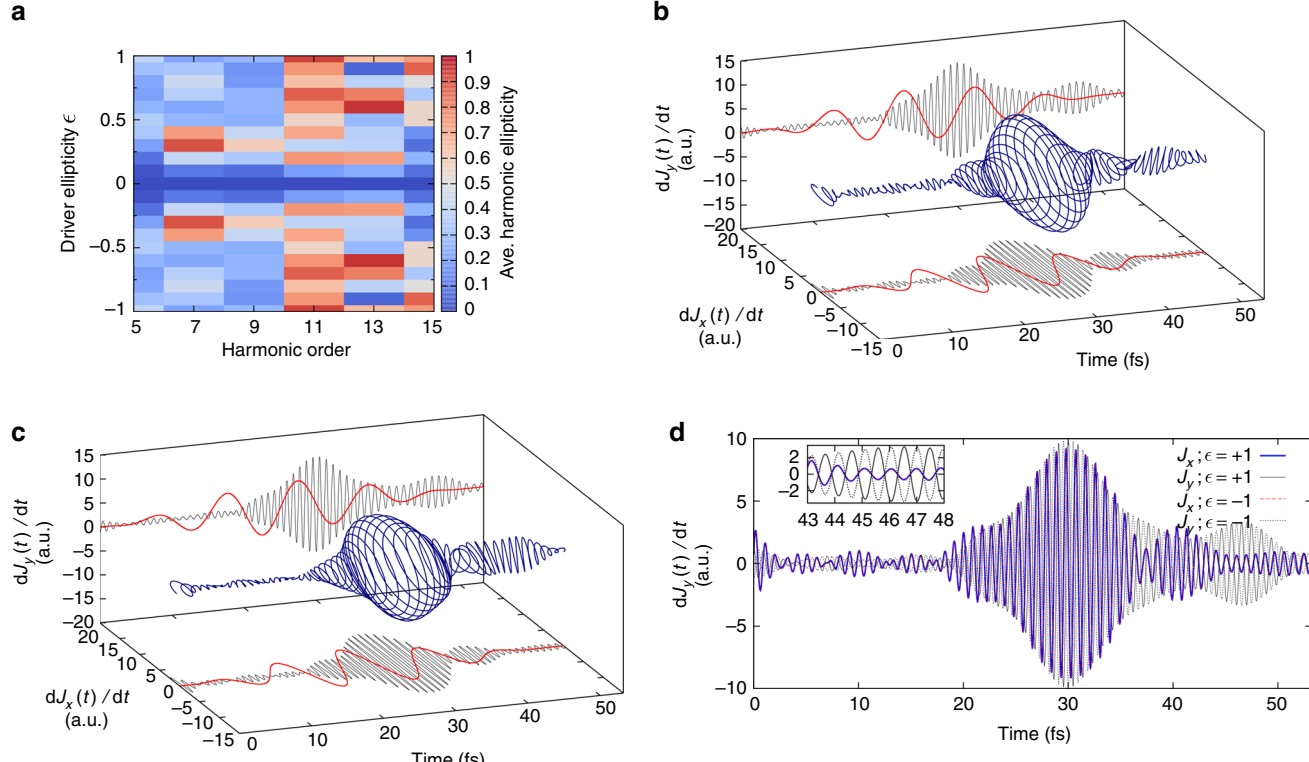

**Fig. 7** Ellipticity of the emitted harmonics for bulk silicon. **a** Same as Fig. 6a but for bulk silicon. **b, c** Evolution of the time-derivative of the electronic current, bandpass filtered around the 11th harmonic for left-handed (**b**) and right-handed (**c**) circular polarization ($\epsilon = \pm 1$). **d** Comparison of the left-handed and right-handed driven 11th harmonic. The x-component is found to be identical for the two cases, whereas the y- components have opposite phase, showing that they have flipped helicity

counter-rotating driver fields[18, 19]. This can be understood based on the following intuitive and simple argument for cubic crystals: the contribution to the emitted current from processes involving $n$ photons appears as the $n$-th power of the incoming electric field (circularly polarized in the $x-y$ plane) $\mathbf{E} = \sqrt{I_0}(1, e^{i\pi/2}, 0)$, i.e., $\sqrt{I_0^n}(1, e^{in\pi/2}, 0)$. Therefore, all $n = (3 + 4l)$-th harmonics will be polarized with the same helicity, and all the $n = (5 + 4l)$-th harmonics with the opposite helicity. Higher-order processes, including absorption and emission of photons, will also result in contributions to the $n$-th order harmonic with the same phase. We note that a more careful analysis, taking into account the crystal's point symmetry group, leads to selection rules of harmonics generated by a circularly polarized driver field[53]. Our results agree perfectly with these selection rules.

We also evaluated the average ellipticity of the emitted harmonics in the case of silicon (see Fig. 7a). Interestingly, we recover the same classification of harmonics obtained from the ellipticity profiles, but here according to the ellipticity of the emitted harmonics. Our results therefore indicate that the generation of harmonics by the interband and intraband mechanisms might lead to distinct ellipticity for the emitted harmonics. Overall, this is yet another proof that interband and intraband mechanisms are affected differently by the ellipticity of the driving field.

Finally, we investigated the possibility of controlling the helicity of the emitted high-order harmonics. We considered in particular the case of the 11th harmonic from bulk Si, shown in Fig. 7b, c, respectively, for left- and right-handed circularly polarized driver pulses. Our results, shown in Fig. 7d, clearly demonstrate the possibility of controlling and flipping the helicity of the emitted harmonics by changing the helicity of the driver field.

## Discussion

In summary, we have investigated the role of ellipticity of the driving laser field on HHG from solids. We have shown that the harmonics of different orders are not affected equally by the ellipticity, and that the symmetries of the Brillouin zone are reflected in the ellipticity profiles. This can be explained by the fact that the interband and intraband mechanisms exhibit a different ellipticity dependence. Moreover, we found that the energy cutoff of the HHG spectra can be strongly modified and even increased when changing the ellipticity of the driving field. Based on our results, we propose to custom-tailor and enhance the HHG in solids by driving the electrons inside the potential energy landscape into specific regions of the Brillouin zone, in particular using laser fields with a temporally evolving polarization state using modern spatial light modulator technology. Finally, we have demonstrated in Si and MgO the possibility of generating (nearly) circularly polarized high-order harmonics with alternating helicities from a single-color circularly polarized laser field, and to control the helicity of these harmonics. Our results pave the way toward ellipticity-based experimental techniques based on high-order harmonic generation, in which solids will play the predominant role.

## Methods

**TDDFT simulations**. The evolution of the wavefunctions and the evaluation of the time-dependent current is computed by propagating the Kohn–Sham equations within TDDFT, as provided by the Octopus package[54], in the adiabatic local-density approximation (LDA)[55]. We employ norm-conserving pseudo-potentials.

It is important to note that within LDA, the bandgap of semiconductors and insulators is underestimated and do not account for excitonic effects[55]. We dealt with this bandgap issue by rescaling the driver intensity in the case of MgO, as explained below, in order to be able to compare to the experimental results of ref. [36].

As shown in ref. [43], dynamical correlations do not affect the HHG spectra of silicon. We checked that the same is true for MgO, for which only harmonics 11 and 13 are modified by dynamical correlations (see Supplementary Note 1 and Fig. 1). As the excitonic effects in these two materials mainly come from the long-range part of the exchange-correlation potential[56], i.e., a renormalization of the Hartree term (which does not play any role in the HHG from Si[43]), excitonic effects are not expected to modify the HHG spectra of materials such as silicon or MgO. We note that this is not necessarily true for all materials, in particular materials with strongly localized excitons, for which bound states will form in the bandgap, or in strongly correlated materials. However, the conclusions of the present work on the role of ellipticity to control HHG would remain valid.

The HHG spectrum is directly obtained from the total electronic current $\mathbf{j}(\mathbf{r}, t)$ as:

$$\mathrm{HHG}(\omega) = \left| \mathrm{FT}\left( \frac{\partial}{\partial t} \int d^3 \mathbf{r} \mathbf{j}(\mathbf{r}, t) \right) \right|^2, \quad (5)$$

where FT denotes the Fourier transform.

**Simulations of HHG from bulk silicon**. All calculations for bulk silicon were performed using the primitive cell of bulk silicon, using a real-space spacing of 0.484 atomic units. We consider a laser pulse of 25-fs duration at full-width half-maximum (FWHM) with a sin-square envelope, and the carrier wavelength $\lambda$ is 3000 nm, corresponding to $\omega = 0.43$ eV. Except for the calculation of the electron dynamics (see below), we employed an optimized $28 \times 28 \times 28$ grid shifted four times to sample the BZ, and we used $I_0 = 10^{11}$ W cm$^{-2}$ (corresponding to a peak intensity in matter of $3.4 \times 10^{11}$ W cm$^{-2}$ for an optical index $n$ of ~3.4). We use the experimental lattice constant $a$ leading to a LDA bandgap of silicon of 2.58 eV. In all our calculations, we used a CEP of $\phi = 0$. We checked (see Supplementary Note 2 and Fig. 2) that the CEP has almost no effect on the ellipticity dependence of HHG in solids for the pulse duration considered here.

**Simulations of HHG from bulk magnesium oxide**. We also performed calculations for bulk MgO, which has a rock-salt crystal structure. We used a real-space spacing of 0.3 atomic units and an optimized $28 \times 28 \times 28$ grid shifted four times to sample the BZ. We use a carrier wavelength $\lambda$ of 1333 nm, corresponding to a carrier photon energy of 0.93 eV to match the experimental conditions used in ref. [36]. The experimental peak intensity in vacuum is ~$10^{13}$ W cm$^{-2}$. The corresponding transmitted peak intensity in matter is therefore ~$9 \times 10^{12}$ W cm$^{-2}$, taking the experimental refractive index of bulk MgO as 1.7175[57], for the considered wavelength. We note that within the LDA, the bandgap of MgO is found to be $E_g^{\mathrm{LDA}} = 4.72$ eV, which strongly underestimates the experimentally observed bandgap of $E_g^{\mathrm{exp}} = 7.83$ eV[58]. Therefore, we use $I_0 = 3 \times 10^{12}$ W cm$^{-2}$ (corresponding to a peak intensity in matter of $5.3 \times 10^{12}$ W cm$^{-2}$ for a refractive index $n = 1.7175$), in order to generate a similar number of harmonics as measured experimentally, to allow a comparison. This value was taken such that the ratio of the bandgap over the intensity in matter is preserved, as this leads, for instance, to the same adiabaticity parameter[59], i.e., the same multi-photon ionization or tunneling regime. In ref. [36], authors used a 50-fs FWHM laser pulse. In our simulations, we used instead a shorter laser pulse of 25-fs FWHM, in order to make the calculations numerically tractable. We found (see Supplementary Note 3 and Fig. 3) that the HHG spectra from bulk MgO are very similar for both 25-fs and 50-fs pulse durations.

**Definition of the harmonic yield**. For each odd harmonic, we define the harmonic yield by integrating the HHG spectrum over the energy region defined by the two neighboring even harmonics, such as the harmonic yield of the $n$-th (odd) harmonic is given by:

$$I_{\mathrm{HH},i}(n) = \int_{(n-1)\omega}^{(n+1)\omega} \mathrm{HHG}_i(\omega') \mathrm{d}\omega', \quad (6)$$

where $\omega$ is the frequency of the laser field. If specified, the subscript $i$ indicates that the yield is computed by only taking into account the $i$-component ($i = x, y$) of the total electronic current.

**Subcycle dynamics of the excited electrons in momentum space**. The simulations of the subcycle dynamics of the excited electrons in momentum space were performed for an intensity of the laser of $I_0 = 5 \times 10^{11}$ W cm$^{-2}$. The ellipticity is taken as $\epsilon = 0.1$. The $t = 0$ time corresponds to the switch-on of the laser pulse. In these simulations, we employed a $27 \times 27 \times 27$ k-point grid, shifted four times, to get the $k_z = 0$ plane in our k-point grid. The total number of excited electrons is defined by projecting the time-evolved wavefunctions ($|\psi_n(t)\rangle$) on the basis of the ground-state wavefunctions $\left( |\psi_{n'}^{\mathrm{GS}}(t)\rangle \right)$

$$N_{\mathrm{ex}}(t) = N_e - \frac{1}{N_\mathbf{k}} \sum_{n,n'}^{\mathrm{occ.}} \sum_\mathbf{k}^{\mathrm{BZ}} \left| \langle \psi_{n,\mathbf{k}}(t) | \psi_{n',\mathbf{k}}^{\mathrm{GS}} \rangle \right|^2, \quad (7)$$

where $N_e$ is the total number of electrons in the system, and $N_\mathbf{k}$ is the total number of k-points used to sample the BZ. The sum over the band indices $n$ and $n'$ run over all occupied states. The momentum-resolved excited electron distribution, as shown in Figs. 3 and 4, is defined here as:

$$N_{\mathrm{ex}}(\mathbf{k}; t) = \frac{1}{N_\mathbf{k}} \left( N_e - \sum_{n,n'}^{\mathrm{occ.}} |\langle \psi_{n,\mathbf{k}}(t) | \psi_{n',\mathbf{k}}^{\mathrm{GS}} \rangle|^2 \right). \quad (8)$$

**Average ellipticity of the emitted harmonics**. For the case of the driving field being polarized in the $x$–$y$ plane, with the major axis of the polarization ellipse along the $x$-axis, we define the average (over the pulse duration) ellipticity of $n$-th harmonic as:

$$|\varepsilon(n\omega)| = \left| \frac{\tilde{E}_y(n\omega)}{\tilde{E}_x(n\omega)} \right| = \sqrt{\frac{I_{\mathrm{HH},y}(n\omega)}{I_{\mathrm{HH},x}(n\omega)}}, \quad (9)$$

where $\omega$ is the frequency of the driving field, $\tilde{E}_i(n\omega)$ is the strength of the $n$-th harmonic electric field along the direction $i = x, y$, and $I_{\mathrm{HH},i}(n\omega)$ the harmonic yield (as defined in "Methods" section), directly obtained from the HHG spectra. However, in some cases, one has to assume that the major axis of the polarization ellipse for the emitted harmonics is along the $y$-axis to get an ellipticity between 0 and 1. Therefore, we use:

$$|\varepsilon(n\omega)| = \min\left( \sqrt{\frac{I_{\mathrm{HH},y}(n\omega)}{I_{\mathrm{HH},x}(n\omega)}}, \sqrt{\frac{I_{\mathrm{HH},x}(n\omega)}{I_{\mathrm{HH},y}(n\omega)}} \right) \quad (10)$$

to evaluate the ellipticity of the emitted harmonics. We note that this can only provide an estimate of the ellipticity of the emitted harmonics, as we use here the harmonic yield obtained by integrating the HHG spectra between the two neighboring even harmonics.

**Data availability**. The data that support the findings of this study are available from the corresponding authors on request, and will be deposited on the NoMaD repository.

The OCTOPUS code is available from http://www.octopus-code.org.

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

## Acknowledgements

We acknowledge financial support from the European Research Council (ERC-2015-AdG-694097), COST Action MP1306 (EUSpec). We thank M. Altarelli for very fruitful discussions. N.T.-D. thanks M.J.T. Oliveira for providing some of the pseudopotential files. F.X.K. and O.D.M. thank N. Klemke and Y. Yang for helpful discussions and acknowledge support by the excellence cluster "The Hamburg Centre of Ultrafast Imaging-Structure, Dynamics and Control of Matter at the Atomic Scale" and the priority program QUTIF (SPP1840 SOLSTICE) of the Deutsche Forschungsgemeinschaft.

## Author contributions

N.T.-D. and A.R. conceived and designed the project. N.T.-D. carried out the code implementation and the numerical calculations. O.D.M. and N.T.-D. worked out the models and the comparison with experiments. N.T.-D., O.D.M., F.X.K. and A.R. participated in the discussion of the results and contributed to the manuscript.

## Additional information

**Competing interests:** The authors declare no competing financial interests.

**Change history:** A correction to this article has been published and is linked from the HTML version of this article.

