## [Peer Review File · Nature Communications]

Reviewers' comments:

Reviewer #1 (Remarks to the Author):

In this manuscript, the authors study using first-principles simulations the role of ellipticity of the driving laser field on high-order harmonic generation in solids. The authors have shown that the symmetries of the Brillouin zone have different effects on different harmonic orders. They have also shown that the harmonic cutoff could be modified when changing the ellipticity of the driving field, and the possibility of generating nearly circularly polarized harmonics with a single-colour polarized laser field.

High-order harmonics from solids is a hot topic, which has gained considerable attention from the scientific community. The current manuscript tries to shed light on the detailed physics involved in past observations, as well as showing the possibility of new phenomenon (such as the generation of circularly polarized harmonics). However, this reviewer finds that in many cases, the explanation of the physics involved is not adequate to merit publication in Nature Communications.

a) For example, in p. 12, last paragraph, the authors attempt to explain the mechanism of the cutoff extension observed in Fig. 2. While a simple explanation, such as the three-step model, may not be possible in such a complex system, the authors need to make better efforts to explain the physics involved in the extension of the cutoff energy in these cases.

b) Another example could be found from p. 13 to 14, where the authors show that elliptically polarized harmonics could be generated using a single-colour elliptically polarized driving field. However, again the mechanism behind the generation of such elliptically polarized harmonics in solids is not clear, and this must be explained more clearly.

Other points that this reviewer noticed:

c) As explained in p. 4, the authors do not take into account dephasing or propagation effects in their simulations, but still find good agreement between their simulation and experiments with bulk MgO. However, to truly show the reliability of their theoretical description, the authors need to at least make an attempt to evaluate if dephasing and propagation effects are negligible under these conditions. I believe even a crude estimation would strongly back this claim.

d) In Fig. 5, the authors claim that there is an extension of the cutoff from the 19th order for $\epsilon = 0$, to the 25th order for $\epsilon = 0.65$. However, the spectra are very noisy in this regime, and it is difficult for this reviewer to evaluate the position of the cutoff. The authors should discuss how they determined the cutoff energies, and if possible, to make it clearer in Fig. 2.

Reviewer #2 (Remarks to the Author):

This paper presents a computational study of high-harmonic generation (HHG) in periodic solids driven by elliptically polarized intense laser fields. HHG in atoms and molecules in the gas phase is a mature field of research with many applications; in recent years, HHG in solids has become of growing interest because there are promising new applications in ultrafast spectroscopy that are only possible using a solid medium.

From the theoretical side, HHG in the gas phase is very well understood, using the standard three-step recollision model. However, this model is not applicable to solids (at least, not in an obvious

manner). There is an intense effort underway to develop new models and computational approaches to describe HHG and other strong-field processes in solids. Some of the recent models are based on simple two-band systems, using the semiconductor Bloch equations, or on semiclassical, trajectory-based approaches. Each of these models has had some degree of success, but there are also discrepancies with experimental observations. A better theory, based on first principles, is clearly needed in this important and ``hot' field of research.

In this paper, such a theory is reported and successfully applied. The work is based on TDDFT, carrying out a full time propagation of the time-dependent Kohn-Sham equations in a periodic solid, driven by an elliptically polarized, time-dependent vector potential corresponding to a femtosecond laser pulse. The authors, a team from the MPI Hamburg, have developed this unique capability, utilizing the octopus code for periodic systems. This is an extremely impressive and important breakthrough in this field.

The authors study two systems, bulk Si and MgO, under the influence of short intense laser pulses with varying degrees of ellipticity. They study the yield of individual harmonics, and discover that the lower harmonics behave quite differently than the higher ones, which suggests different mechanisms. The paper presents a careful analysis in terms of inter- and intraband dynamics, and concludes that there is a rather complex interplay that could be exploited to control HHG from solids. Indeed, it is found that a finite ellipticity can increase the extent of the HHG plateau and can give rise to elliptically polarized harmonics. These findings are extremely promising for future applications.

Overall, this is an interesting and important paper, which deserves to be published in Nature Communications. However, I have a few comments which the authors should address.

1. In Section III.C, it is written than MgO has zinc-blende crystal structure. This is not correct: MnO has Rocksalt structure. If it were zincblende, then inversion symmetry would be broken, and even harmonics would show up.
2. In Eq. (1), the ellipticity parameter ϵ is not defined. It should be defined here, and its range $-1 < \epsilon < +1$ should be specified.
3. In Fig. 2 it is not said how the angle θ is measured. I assume that it is defined with respect to the Gamma-X axis, like in Fig. 1 for silicon. But it should be said explicitly.
4. The main emphasis of this paper is on unraveling the interplay between intra - and interband dynamics. However, the reader has a hard time understanding the difference between the two, until about the second half of the paper (section I.C). I think that the paper would benefit from some rearrangement. The authors should explain early on that by interband they simply mean transitions from the valence to the conduction band, similar to bound-bound transitions in atoms; by intraband dynamics they mean that an electron wavepacket that is excited into the conduction band will be driven by the field, thus exploring the complex band structure in k -space. This is, in a sense, similar to the second step in the recollision model, where the ionized electron is driven by the field (although here, of course, there is no ionization). Without clearly defining these concepts upfront, much of the discussion in the first half of the paper is hard to understand.
5. In Section I.B, the authors discuss electronic excitations in terms of tunneling into the conduction band. This is also brought up one page later, right before Eq. (3), where interband Zener tunneling is said to occur. I wonder why there is no mention of multiphoton processes, which, I think, are an equally important way of exciting an electron with photons whose energy is less than the band gap. In atomic strong-field processes, there is a long tradition of discussion multiphoton ionization versus

tunneling or above-the-barrier ionization processes. I think this will also play a role for solids.

6. The TDDFT calculations are done using the ALDA, which is the only numerically feasible approach at present. However, the ALDA has well-known deficiencies. The band gap underestimation will play a role in the different behavior of individual harmonics. This is implicitly accounted for in Fig. 2, where the 19th experimental harmonic is compared with the calculated harmonics 21 and 23. Perhaps this could be stated explicitly. But there is another important weakness of the ALDA, namely, it does not produce excitons. The authors should mention this, and discuss how the inclusion of excitonic effects would affect the inter- and intraband mechanisms for HHG in solids.

Reviewer #3 (Remarks to the Author):

This is an interesting and well written paper.

Investigation of high-harmonic generation (HHG) from solids, both theoretically and experimentally, is a hot topic of modern laser and strong-field physics. This HHG is different than usual HHG from atomic gases. It is explained by inter and intraband models, but realistic study of microscopic HHG in solids is possible using *ab initio* approaches. In the present work, the time-dependent density functional theory was used for this purpose [see the recently published *Phys. Rev. Lett.* 118, 087403 (2017) by the same authors].

New result in the present work is a theoretical analysis of the ellipticity dependence of HHG in solids. It was found that interband and intraband microscopic mechanisms responsible for HHG in solids are affected differently by the driving field ellipticity.

The introductory part is clearly and competently written.

In addition to (or instead of) the unpublished reference 26, the reference *Phys. Rev. A* 93, 051402(R) (2016), in which the attospin was introduced for the first time should be mentioned.

The main body of the paper is also well written and illustrated by numerical examples and Supplementary videos. The Method section is informative.

Having all this in mind, I recommend publication of this paper.

Reviewers' comments:

Reviewer #1 (Remarks to the Author):

In this manuscript, the authors study using first-principles simulations the role of ellipticity of the driving laser field on high-order harmonic generation in solids. The authors have shown that the symmetries of the Brillouin zone have different effects on different harmonic orders. They have also shown that the harmonic cutoff could be modified when changing the ellipticity of the driving field, and the possibility of generating nearly circularly polarized harmonics with a single-colour polarized laser field.

We thank the referee for summarizing the main findings of our work.

High-order harmonics from solids is a hot topic, which has gained considerable attention from the scientific community. The current manuscript tries to shed light on the detailed physics involved in past observations, as well as showing the possibility of new phenomenon (such as the generation of circularly polarized harmonics). However, this reviewer finds that in many cases, the explanation of the physics involved is not adequate to merit publication in Nature Communications.

We have addressed below in detail all the points raised by the referee.

a) For example, in p. 12, last paragraph, the authors attempt to explain the mechanism of the cutoff extension observed in Fig. 2. While a simple explanation, such as the three-step model, may not be possible in such a complex system, the authors need to make better efforts to explain the physics involved in the extension of the cutoff energy in these cases.

We agree with the referee that this point was not clearly explained in the previous version of the manuscript. We have therefore modified our manuscript to clarify and explain the underlying physics:

“The position in momentum space k of the electron wavepacket determines its energy, its velocity (via the band dispersion) and its coupling to other bands. As shown in Ref.43, when the JDOS taken from the region of the BZ explored by the electrons is low, the harmonic yield and the cutoff are increased. The cutoff extension can be understood as resulting from the lower JDOS seen by electrons when driven with a finite ellipticity laser, which implies a lower contribution of the interband mechanism in favor of the intraband mechanism, compared to the linearly polarized case.”

b) Another example could be found from p. 13 to 14, where the authors show that elliptically polarized harmonics could be generated using a single-colour elliptically polarized driving field. However, again the mechanism behind the generation of such elliptically polarized harmonics in solids is not clear, and this must be explained more clearly.

We thank the referee for pointing out that, apart from providing unambiguous proof by the numerical results, in the originally submitted manuscript we did not provide an intuitive picture to explain the underlying mechanism responsible for the generation of elliptically polarized harmonics in solids.

However, we want to point out that the generation of elliptically polarized harmonics from elliptically polarized driver in solids (or liquid crystals) is actually a very old topic in nonlinear optics (see for instance [Phys. Rev. 171, 1104 (1968)], [Phys. Rev. Lett. 25, 23 (1970)]).

The selection rules for harmonics from a circularly polarized driver field was predicted and explained many years ago. In particular, “Selection rules for circularly polarized waves in nonlinear Optics” [Tang and Rabin, Phys. Rev. B 3, 4025 (1970)] explains, based only on group theory, what are the harmonics' polarization and helicity that should be obtained from a circularly polarized driver.

Our numerical findings are in full agreement with their theoretical results, especially the alternating helicity of the circularly polarized harmonics in Si or MgO.

We have modified our manuscript to clarify this point.

Other points that this reviewer noticed:

c) As explained in p. 4, the authors do not take into account dephasing or propagation effects in their simulations, but still find good agreement between their simulation and experiments with bulk MgO. However, to truly show the reliability of their theoretical description, the authors need to at least make an attempt to evaluate if dephasing and propagation effects are negligible under these conditions. I believe even a crude estimation would strongly back this claim.

We thank the referee for this important remark. We are happy that this reviewer appreciates the “good agreement between (our) simulation and experiments”. Light propagation and dephasing effects are indeed important effects.

However, we reported in Fig. 2 a comparison of the harmonic yield with respect to the driver ellipticity. Propagation and dephasing effects, however, are affecting the spectra equally for all the driver ellipticities. Therefore, while the absolute value of the ellipticity profile will be affected by these effects, the lineshape of this profile will remain unaffected, thus explaining the observed good agreement with experiment.

To be more precise, MgO is a cubic isotropic material. Therefore the linear absorption by MgO only depends on the wavelength and not on the polarization state of the harmonic.

For the harmonics being emitted from the last few nanometers near the back surface of the crystal, nonlinear effects in the propagation of the harmonics can be neglected.

Concerning the dephasing effects, which originate from electron-electron and electron-phonon scattering, if they are treated within the relaxation time approximation (T_2), the dephasing time is a material property, which is independent of the excitation condition. This is the approximation which has been adopted in most of the HHG papers, even producing a good agreement with experiments [45-48]. Therefore, assuming a constant dephasing time T_2 , dephasing will not change the lineshape of the ellipticity profiles, as for the light-propagation effects.

Electron-phonon scattering is not expected to play a relevant role as we considered a short laser pulse.

We added the following discussion in our manuscript:

“It is important to note that even if propagation and effects are not included here, we still obtain a good agreement with experiments. Propagation effects can affect the harmonic spectra and the harmonic yield. However, as MgO is a cubic isotropic material, the linear absorption only depends on the wavelength but not on the polarization state of the emitted harmonics. Therefore, the harmonic yield of the different polarizations will be affected equally by light propagation

effects, irrespective of the driver's and harmonics' ellipticities. Moreover, since the harmonics originate within few nanometers close to the back surface of the crystal, nonlinear effects in propagation of the emitted harmonics can be neglected.

Concerning the dephasing effects, which originate from electron-electron and electron-phonon scattering, if they are treated within the relaxation time approximation (T_2), the dephasing time is a material property, assumed to be independent of the excitation condition. This is the approximation which has been adopted in most of the HHG papers, even producing a good agreement with experiments [45-48]. Most importantly, assuming a constant dephasing time T_2 , dephasing will not change the lineshape of the ellipticity profiles."

d) In Fig. 5, the authors claim that there is an extension of the cutoff from the 19th order for $\epsilon = 0$, to the 25th order for $\epsilon = 0.65$. However, the spectra are very noisy in this regime, and it is difficult for this reviewer to evaluate the position of the cutoff. The authors should discuss how they determined the cutoff energies, and if possible, to make it clearer in Fig. 2.

We agree with the referee that the spectra look noisy. We defined the energy cutoff as the highest well defined harmonic peak (above numerical background). Even if this definition is not very precise, we want to stress here that our main finding is that new harmonic peaks emerge at higher photon energy in the harmonic spectra, compared to the linear polarization case, thus extending the energy cutoff. This does not depend on the exact definition of the cutoff position.

We modified Fig. 5 to include an inset showing a zoom into the spectral region around the energy cutoff.

We hope that referee #1 finds the above explanations and manuscript revisions adequate such that the revised manuscript warrants publication in Nature Communications.

Reviewer #2 (Remarks to the Author):

This paper presents a computational study of high-harmonic generation (HHG) in periodic solids driven by elliptically polarized intense laser fields. HHG in atoms and molecules in the gas phase is a mature field of research with many applications; in recent years, HHG in solids has become of growing interest because there are promising new applications in ultrafast spectroscopy that are only possible using a solid medium.

From the theoretical side, HHG in the gas phase is very well understood, using the standard three-step recollision model. However, this model is not applicable to solids (at least, not in an obvious manner). There is an intense effort underway to develop new models and computational approaches to describe HHG and other strong-field processes in solids. Some of the recent models are based on simple two-band systems, using the semiconductor Bloch equations, or on semiclassical, trajectory-based approaches. Each of these models has had some degree of success, but there are also discrepancies with experimental observations. A better theory, based on first principles, is clearly needed in this important and "hot" field of research.

In this paper, such a theory is reported and successfully applied. The work is based on TDDFT, carrying out a full time propagation of the time-dependent Kohn-Sham equations in a periodic solid, driven by an elliptically polarized, time-dependent vector potential corresponding to a femtosecond laser pulse. The authors, a team from the MPI Hamburg, have developed this unique capability, utilizing the octopus code for periodic systems. This is an extremely impressive and important breakthrough in this field.

The authors study two systems, bulk Si and MgO, under the influence of short intense laser pulses with varying degrees of ellipticity. They study the yield of individual harmonics, and discover that the lower harmonics behave quite differently than the higher ones, which suggests different mechanisms. The paper presents a careful analysis in terms of inter- and intraband dynamics, and concludes that there is a rather complex interplay that could be exploited to control HHG from solids. Indeed, it is found that a finite ellipticity can increase the extent of the HHG plateau and can give rise to elliptically polarized harmonics. These findings are extremely promising for future applications.

Overall, this is an interesting and important paper, which deserves to be published in Nature Communications. However, I have a few comments which the authors should address.

We thank the referee for the very positive evaluation of our work and the recommendation for publication.

1. In Section III.C, it is written than MgO has zinc-blende crystal structure. This is not correct: MnO has Rocksalt structure. If it were zincblende, then inversion symmetry would be broken, and even harmonics would show up.

We thank the referee for pointing out this error. Indeed MgO has rock-salt structure. We corrected the manuscript accordingly.

2. In Eq. (1), the ellipticity parameter epsilon is not defined. It should be defined here, and its range $-1 < \text{epsilon} < +1$ should be specified.

We have modified the text to define epsilon and its range:

“In the following, the ellipticity parameter is denoted as epsilon, which varies from -1 (left-handed circular polarization) to 0 (linear polarization) to +1 (right-handed circular polarization).”

3. In Fig. 2 it is not said how the angle theta is measured. I assume that it is defined with respect to the Gamma-X axis, like in Fig. 1 for silicon. But it should be said explicitly.

We thank the referee for his/her good remark. The angle is indeed defined as in Fig.1. We clarified this point by adding the following sentence in the caption of Fig.2
“As in Fig.1, theta=0deg. corresponds to the ΓX direction.”

4. The main emphasis of this paper is on unraveling the interplay between intra- and interband dynamics. However, the reader has a hard time understanding the difference between the two, until about the second half of the paper (section I.C). I think that the paper would benefit from some rearrangement. The authors should explain early on that by interband they simply mean transitions from the valence to the conduction band, similar to bound-bound transitions in atoms; by intraband dynamics they mean that an electron wavepacket that is excited into the conduction band will be driven by the field, thus exploring the complex band structure in k-space. This is, in a sense, similar to the second step in the recollision model, where the ionized electron is driven by the field (although here, of course, there is no ionization). Without clearly defining these concepts upfront, much of the discussion in the first half of the paper is hard to understand.

We understand the concern of the referee and therefore have modified the text to improve its readability.

To improve our manuscript, we added the following sentence in Sec. I.A:

“HHG from solids originates from two mechanisms, the interband mechanism, which corresponds to direct electron-hole recombination, and the intraband dynamics, in which the carriers are accelerated within the bands by the driver field.”

5. In Section I.B, the authors discuss electronic excitations in terms of tunneling into the conduction band. This is also brought up one page later, right before Eq. (3), where interband Zener tunneling is said to occur. I wonder why there is no mention of multiphoton processes, which, I think, are an equally important way of exciting an electron with photons whose energy is less than the band gap. In atomic strong-field processes, there is a long tradition of discussion multiphoton ionization versus tunneling or above-the-barrier ionization processes. I think this will also play a role for solids.

We thank the referee for his/her good remark, which helped us to improve the discussion of our results.

It is indeed true that both tunneling and multiphoton ionization play important role in the HHG process in solids.

We have slightly modified the text to not restrict our discussion to tunneling only, as our theoretical model encompasses both excitation mechanisms.

6. The TDDFT calculations are done using the ALDA, which is the only numerically feasible approach at present. However, the ALDA has well-known deficiencies. The band gap underestimation will play a role in the different behavior of individual harmonics. This is implicitly accounted for in Fig. 2, where the 19th experimental harmonic is compared with the calculated harmonics 21 and 23. Perhaps this could be stated explicitly. But there is another important weakness of the ALDA, namely, it does not produce excitons. The authors should mention this, and discuss how the inclusion of excitonic effects would affect the inter- and intraband mechanisms for HHG in solids.

The referee is indeed right to point out that the ALDA has well known failures.

The underestimation of the band-gap was in fact already discussed in the originally submitted manuscript, in the Method section:

“We note that within the local-density approximation (LDA), the band gap of MgO is found to be $E_g=4.72$ eV, which strongly underestimates the experimentally observed band gap of $E_g=7.83$ eV. Therefore, we use $I_0=3\times 10^{12}$ W.cm⁻² corresponding to a peak intensity in matter of 5.3×10^{12} W.cm⁻² for $n=1.7175$), in order to generate a similar number of harmonics as measured experimentally, to allow a comparison.”

Following the referee's suggestion, we have modified the text (see Section method) to state this point more explicitly:

“It is important to note that within LDA, the band gap of semiconductors and insulators is underestimated and do not account for excitonic effects.”

We also added there the following discussion :

“We dealt with this band-gap issue by rescaling the driver intensity in the case of MgO, as explained below, in order to be able to compare to the experimental results of Ref.36.

As shown in Ref.43, dynamical correlations do not affect the HHG spectra of silicon.

We checked that the same is true for MgO, for which only harmonic 11 and 13 are modified by dynamical correlations (see Supplementary Information). As the excitonic effects in these two materials mainly come from the long-range part of the exchange-correlation potential,⁵⁸ i.e., a renormalization of the Hartree term (which does not play any role in the HHG from Si⁴³), excitonic effects are not expected to modify the HHG spectra of materials such as silicon or MgO. We note that this is not necessarily true for all materials, in particular materials with strongly localized excitons, for which bound states will form in the band gap, or in strongly correlated materials.

However, the conclusions of the present work on the role of ellipticity to control HHG would remain valid.”

and the following sentence to explain the rescaling of the laser intensity

“This value was taken such that the ratio of the band-gap over the intensity in matter is preserved, as this leads, for instance, to the same adiabaticity parameter⁶¹, i.e., the same multiphoton ionization or tunneling regime.”

We thank referee #2 again for the very positive opinion on our work and the recommendation for publication. We hope that the manuscript revisions and explanations give above have now adequately addressed all points raised by this referee.

Reviewer #3 (Remarks to the Author):

This is an interesting and well written paper.

We thank referee #3 for this very positive remark.

Investigation of high-harmonic generation (HHG) from solids, both theoretically and experimentally, is a hot topic of modern laser and strong-field physics. This HHG is different than usual HHG from atomic gases. It is explained by inter and intraband models, but realistic study of microscopic HHG in solids is possible using ab initio approaches. In the present work, the time-dependent density functional theory was used for this purpose [see the recently published Phys. Rev. Lett. 118, 087403 (2017) by the same authors].

New result in the present work is a theoretical analysis of the ellipticity dependence of HHG in solids. It was found that interband and intraband microscopic mechanisms responsible for HHG in solids are affected differently by the driving field ellipticity.

The introductory part is clearly and competently written.

We thank the referee for this positive evaluation.

In addition to (or instead of) the unpublished reference 26, the reference Phys. Rev. A 93, 051402(R) (2016), in which the attospin was introduced for the first time should be mentioned.

The main body of the paper is also well written and illustrated by numerical examples and Supplementary videos. The Method section is informative.

We thank the referee for his/her positive opinion on our work.

Having all this in mind, I recommend publication of this paper.

In summary, we want to thank the referee for her/his very positive analysis of our work. We have included the reference [Phys. Rev. A 93, 051402(R) (2016)] as new reference 27 in the revised manuscript and hope that our manuscript is now ready for publication.

REVIEWERS' COMMENTS:

Reviewer #1 (Remarks to the Author):

The authors have responded to all of the questions that this reviewer had, and thus have considerably improved the manuscript. This reviewer recommends the publication of this manuscript in Nature Communications.

Reviewer #2 (Remarks to the Author):

The authors have responded to the comments by the referees. I'm satisfied with their responses, and the changes and clarifications made in the manuscript: the paper is now very clear, and the physical mechanisms are well explained. I recommend publication.